# Construction of a Ternary Composite Colloidal Structure of Zein/Soy Protein Isolate/Sodium Carboxymethyl Cellulose to Deliver Curcumin and Improve Its Bioavailability

**DOI:** 10.3390/foods12142692

**Published:** 2023-07-13

**Authors:** Chong Yu, Jingyu Shan, Hao Ju, Xiao Chen, Guangsen Xu, Yanchao Wu

**Affiliations:** School of Marine Science and Technology, Harbin Institute of Technology, Weihai 264209, China; yc15776578163@163.com (C.Y.); sjy20040721@163.com (J.S.); juhao17@mails.ucas.ac.cn (H.J.); wm13613639977@163.com (X.C.); shi.heng2009@163.com (G.X.)

**Keywords:** zein, soy protein isolate, CMC-Na, curcumin, nanoparticle

## Abstract

This work presents the fabrication of ternary nanoparticles (Z/S/C NPs) comprising zein (Z), soy protein isolate (SPI) and carboxymethylcellulose sodium (CMC-Na) through a pH-driven method. The results showed that the smallest particle size (71.41 nm) and the most stable zeta potential, measuring −49.97 mV, were achieved with the following ratio of ternary nanoparticles Z/SPI/CMC-Na (2:3:3). The surface morphology of the nanoparticles was further analyzed using transmission electron microscopy, and the synthesized nanoparticles were utilized to encapsulate curcumin (Cur), a hydrophobic, bioactive compound. The nanoparticles were characterized using a particle size analyzer, infrared spectroscopy, and X-ray diffraction (XRD) techniques. The results revealed that the formation of nanoparticles and the encapsulation of Cur were driven by electrostatic, hydrogen-bonding and hydrophobic interactions. The drug loading efficiency (EE%) of Z/S/C-cur nanoparticles reached 90.90%. The Z/S/C ternary nanoparticles demonstrated enhanced storage stability, photostability and simulated the gastrointestinal digestion of Cur. The release of Cur and variations in the particle size of nanoparticles were investigated across different stages of digestion. The biocompatibility of the Z/S/C ternary nanoparticles was assessed by conducting cell viability assays on HepG2 and L-O2 cells, which showed no signs of cytotoxicity. These results suggested that the ternary composite nanoparticles have potential in delivering nutritional foods and health-promoting bioactive substances.

## 1. Introduction

Curcumin (Cur) is a kind of polyphenol, primarily obtained from turmeric. It is well known for its vivid color and is extensively utilized for food coloring and industrial pigment applications [1,2]. As a type of substance extracted from plants, people have discovered, through continuous development and exploration, that it can be used as a spice herb and for many other applications [3]. As science and technology continue to advance and our understanding of biomedicine deepens, food pigments like Cur have resurfaced with a renewed perspective.

Researchers have gradually uncovered the exceptional pharmacological properties of Cur, which include anti-inflammatory, antioxidant, antitumor, antibacterial, antiviral and additional beneficial effects [4,5,6]. This highlights the significant potential of Cur in preventing and control various human ailments, thereby promoting human health and well-being. Despite Cur being widely versatile and bioactive, its low insolubility in water, stringent storage requirements and low bioavailability can render it unsuitable for various therapeutic and biotechnology functions [7]. 

To overcome the existing limitations of Cur, researchers are continually seeking novel delivery methods. By utilizing Cur as a model drug for oral and in vivo delivery, researchers aimed to enhance its water solubility, maintain its biological activity, and protect it from fluctuations in the external acid–base environment. These novel delivery vehicles include liposomes [8], nanoemulsions [9], electrospray [10], electrospinning [11,12] and protein carriers (such as bovine serum albumin nanoparticle). The application of animal protein nanoparticles has been limited due to allergies, religion, living, high cost and eating habits [13]. Hence, progress of plant protein carriers has become a trend because of its excellent fitness and cost-effectiveness [14]. Currently, researchers have developed varied plant protein carriers on the basis of zein [15], millet prolamin [16] or SPI [17,18] to deliver hydrophobic active substances.

Zein is an alcohol-soluble protein that is extracted from corn. It is known for its unique amino acid sequence, which results in its poor water solubility. However, it can be dissolved in a high-concentration ethanol solution and alkaline solution with a pH greater than 11 [19]. As a class of hydrophobic molecules, zein exerts uniquely high hydrophobicity, which increases its tendency to aggregate in water.

Zein can self-assemble into a granular form in water and can be maintained at the nanometer level [20,21]. At the same time, the interior of this spherical nanocarrier is hydrophobic, and the exterior is hydrophilic, so it is amphiphilic. The results of research have indicated that zein has a potential to be used to prepare a plant protein nanoparticle. Zein can be dissolved in ethanol and form nanoparticles under certain conditions, using the anti-solvent precipitation method [22]. Due to its low cost, ease of acquisition, and convenient preparation, zein is considered a promising nanocarrier for delivering bioactive substances. In a prior investigation [23], researchers coated tremella polysaccharide onto the surface of zein nanoparticles using the anti-solvent precipitation method. The resulting zein–tremella polysaccharide nanoparticles exhibited excellent physical and chemical stability, ultimately enhancing the biological activity of Cur. To improve the utilization of Cur, zein was used as a carrier with propylene glycol alginate and rhamnolipid dissolved in ethanol using the anti-solvent precipitation method, and the mixture was then added into the water phase. During the rotary evaporation process of the ethanol, a complex of nanoparticles was formed [24,25]. However, ethanol is a flammable and explosive substance, which poses transportation challenges for drug delivery. The previous experiments and studies used a high concentration and dose of ethanol. In addition, the high cost of ethanol in industrial production makes the zein nanoparticles prepared by this method unsuitable for commercialization and industrialization. Furthermore, various factors limit their rapid spread and application. For example, as people’s understanding of health concepts deepens, the demand for non-alcoholic food is increasing. Consequently, the application of the method mentioned above is becoming more limited [26].

Recently, a novel method for zein preparation has been developed [27]. The process involves dissolving zein in a highly concentrated alkaline solution and adjusting the pH to neutral or slightly alkaline to induce the loss of protein solubility under severe pH changes, resulting in the formation of hydrophobic structures. These structures form spherical shapes and can encapsulate some drugs that are soluble in alkali and maintain their activity over a short time. Cur is one such drug with poor solubility under acidic and neutral conditions, but which easily deprotonates in sodium hydroxide solution owing to its hydroxyl group [28]. It becomes soluble in an alkaline solution and is thus cosoluble with alkaline corn proteins, allowing it to be loaded into zein nanoparticles during pH changes.

The simple zein nanoparticles are very unstable due to their isoelectric point (pI) of 6.2 [29], and maize prolamin is highly susceptible to precipitation in neutral environments and around the isoelectric point. To improve the stability of zein nanoparticles, researchers have explored different approaches. Firstly, they have examined the effect of pH changes and sought to identify weak acids that can be utilized to adjust the solution’s pH, such as citric acid [30], which can improve the stability of the protein structure. Secondly, various natural substances have been investigated to stabilize zein nanoparticles, including polysaccharides (fucoidan) [22], xanthan gum [31] and pectin [32], as well as proteins such as whey protein [33], pea isolate protein [34] and heterophylla protein [35]. By improving the adaptability of the nanocarriers to the surrounding environment and enhancing the stability of zein, these substances have the potential to increase the versatility and effectiveness of zein nanoparticles as a delivery system for bioactive compounds. In a recent study [36], it was found that the combination of sodium caseinate and fucoidan significantly improved the bioavailability of the drug stabilizer compared to using a single component alone. This combination also resulted in a reduction in particle size, making the nanocarrier more easily absorbed by the human body. Additionally, it broadened the range of pH applications, making the nanocarrier suitable for use in a wider pH environment. Sodium carboxymethyl cellulose (CMC-Na), is a hydrophilic polysaccharide, and a widely used derivative of cellulose in the pharmaceutical industry. As a anionic polymer, CMC-Na is utilized as a carrier for delivering bioactive substances [37].

Our research revealed a lack of data and studies in the existing literature on the combined effect of SPI and CMC-Na related to the zein delivery system. Hence, our study aims to investigate the stability and synergistic effect of SPI and CMC-Na in the preparation of zein nanoparticles using a pH-driven method. In this experiment, we utilized Cur as a model drug for encapsulation in the zein complex nanocarriers. In this study, we investigated the impact of adding SPI and CMC-Na on the particle size, potential, and PDI of zein nanoparticles. We characterized these nanoparticles at the microscopic level using transmission electron microscopy. After successfully constructing zein/SPI/CMC-Na nanocarriers, we used Cur as a model drug to investigate the stability of Z/S/C-cur drug-loaded nanocarriers. We studied the storage stability of Z/S/C-cur, the bioavailability of gastrointestinal digestion and release, pH stability, particle stability and photostability. Furthermore, we evaluated the potential cytotoxicity of the zein/SPI/CMC-Na complex nanocarriers to assess their impact on human health. This study aims to establish a new nanocarrier structure for the delivery of the hydrophobic active substance Cur.

## 2. Materials and Method

### 2.1. Materials

Zein was obtained from Macklin Company, Shanghai, China (purity ≥ 95%). Soy protein isolate (SPI) (purity ≥ 98%), sodium carboxymethyl cellulose (CMC-Na) (purity ≥ 98%) and curcumin (Cur) (purity ≥ 95%) were purchased from Shanghai Energy Company, China. All reagents and solvents used were of analytical grade (from Sinopharm Beijing, China). The solution was used ultrapure water (0.1–0.055 μs/cm) throughout the whole experiment.

### 2.2. Preparation

#### 2.2.1. Preparation of Stock Solution (Zein, SPI, CMC-Na, Cur)

The zein stock solution (concentration (10 mg/mL)) was prepared by dissolving 1 g of zein in a 100 mL of 25 mmol/L NaOH solution. Different qualities SPI (0.5 g, 1 g, 2 g, 3 g, 4 g and 5 g) were dissolved in a 100 mL of 25 mmol/L NaOH solution, and various amounts of CMC-Na (0.75 g, 1.5 g, 3 g, 4.5 g, 6 g and 7.5 g) were also dissolved in a 100 mL of 25 mmol/L NaOH solution. 2 mg/mL Cur was dissolved in a 25 mmol/L NaOH solution. 

#### 2.2.2. Preparation of Zein-Based Nanoparticles

The self-assemble zein nanoparticles were produced by the pH-driven method according to a previously published work with some modifications [38]. The preparations of zein nanoparticles, Z/S nanoparticles and Z/S/C complex nanoparticles were performed as follows:

(1) For the preparation of zein nanoparticles. Firstly, 2 mL of zein stock solution (10 mg/mL) was into 10 mL of 25 mmol/L NaOH solution under 800 rpm stir. 2 mL of zein stock solution was added into different quality ratio SPI stock solution (10 mL) to prepare the Z/S complex nanoparticles. The final ratio of zein to SPI are 4:1, 2:1, 1:1, 2:3, 1:2 and 2:5, respectively. The Z/S/C complex nanoparticles are prepared by stirring the mixture of 2 mL of zein stock solution (10 mg/mL), 10 mL SPI and 10 mL CMC-Na together under 800 rpm stir. Make sure the ratio of SPI to CMC-Na are 4:1, 2:1, 1:1, 1:2 and 2:5, respectively.

(2) After mixing different complex nanoparticles for 10 min, 0.5 mol/L citric acid was used to adjust the pH value of the different complex nanoparticle groups to 7.5. Subsequently, all the nanoparticles were centrifuged under 5000 rpm for 15 min to remove excess insoluble materials. The different mass ratio of the prepared Z, Z/S and Z/S/C nanoparticles are shown in Table 1.

#### 2.2.3. Preparation of Cur-Loaded Nanoparticles

For the preparation of Cur-loaded nanoparticles, a Cur stock solution was added into the nanoparticle complex before pH regulation. Briefly, after adding 2 mL of the Cur stock solution into the nanoparticle complex, the pH values were regulated with citric acid, creating Z-cur, Z/S-cur, and Z/S/C-cur nanoparticles.

### 2.3. Characterization of the Complex Nanoparticles

#### 2.3.1. Particle Size, Polydispersity Index (PDI) and Zeta Potential of Nanoparticles

Characterization of nanoparticles was performed using a Zetasizer Nano-ZS90 (Malvern, Worcestershire, UK). Before the measurement, take 500 μL of each nano-dispersion, prepare up to 5 mL with Ultrapure Water, and dilute it 10 times to avoid inaccurate results due to scattering effects. All data were measured three times and averaged. After the instrument was turned on, it was preheated for 10 min to prevent result deviations caused by environmental factors.

#### 2.3.2. Encapsulation Efficiency (EE%)

Briefly, freshly prepared nanoparticles from each group were centrifuged at 4000 rpm for 10 min to remove insoluble matter and some large particles, and then 0.2 mL of the supernatant of the nanoparticle dispersion was taken and diluted 15 times with dimethyl sulfoxide (DMSO) [39]. After thorough mixing, the absorbance was measured at 426 nm using a UV-gradiometer. The specific EE values were calculated according to the standard curve (*R*^2^ < 0.9998). The remaining nano-dispersion was freeze-dried for subsequent experimental analysis.
(1)$$\mathrm{EE}\;\left(\%\right)=\frac{\left(\mathrm{Total}\;\mathrm{Cur}\;-\;\mathrm{Free}\;\mathrm{Cur}\right)}{\mathrm{Total}\;\mathrm{Cur}}\times 100\;\%$$

#### 2.3.3. Transmission Electron Microscopy (TEM)

A new, bare zein, Z/S and Z/S/C nanoparticle dispersion was prepared for TEM analysis. Using a transmission electron microscope (JEOL-2100 microscope) at 200 kV, 10 μL samples were dropped onto the plasma (glow power generation) carbon film network, using 2% (*w/v*) phosphotungstic acid as Stain, negatively stain Z, Z/S, Z/S/C nanoparticles for one min.

#### 2.3.4. Fourier Transform Infrared Spectroscopy (FTIR)

FTIR spectroscopy of the nanoparticles (Z-cur, Z/S-cur, Z/S/C-cur) and the four single components (zein, SPI, CMC-Na, Cur) was carried out. Spectra were collected from 600 to 4000 cm^−1^ in 32 scans with a resolution of 4 cm^−1^.

#### 2.3.5. XRD

After the nano-dispersion was centrifuged, the supernatant was freeze-dried. Next, the X-ray diffractometer (DX2700, Dandonghaoyuan, Dandong, China) was used to analyze the 7 samples (zein, SPI, CMC-Na, Cur, Z-cur, Z/S-cur, Z/S/C-cur). The data were collected in the range of 3° to 50° (2^θ^) in steps of 0.02°.

### 2.4. Stability of the Nanoparticles

#### 2.4.1. pH Stability

The pH stability of the co-assembled Z/S/C NPs was experimentally determined based on previous work [16]. Briefly, the prepared Z-cur, Z/S-cur, and Z/S/C-cur as well as the pH was adjusted using 1 mol/L HCl and 25 mmol/L NaOH at room temperature. After storage for 6 h, NPs were diluted ten-fold with ultrapure water. The appearance and particle size and zeta potential of the samples were observed.

#### 2.4.2. Ionic Strength Stability

To test the ion strength stability of the nanoparticles, we conducted the following experiments. Firstly, Z-cur and Z/S-cur nanoparticles were mixed with NaCl solutions of different concentrations (0, 21, 42, 63, 84 and 105 mmol/L) in equal volumes. Secondly, Z/S-cur and Z/S/C-cur nanoparticles were mixed with NaCl solutions of different concentrations (0, 50, 100, 150, 200 and 250 mmol/L) in equal volumes. After preparing the two sets of nanoparticle dispersions, the appearance, particle size and zeta potential of each sample were observed after overnight incubation.

#### 2.4.3. Photochemical Stability

In this experiment, the photochemical stability of the three types of nanoparticles (Z-cur, Z/S-cur, Z/S/C-cur) and Cur dissolved in ethanol (as a control) was determined using the method described in [40]. In brief, each group of nanoparticles was irradiated with UV light (30, 60, 90, 120, 150, 180 min), and the remaining content of Cur in the nanoparticles was measured after irradiation.

#### 2.4.4. Storage Stability 

The newly prepared Z-cur, Z/S-cur and Z/S/C-cur nanoparticles were centrifuged and tested at 0, 7 and 14 days. They were stored under refrigeration at 4 °C. The data obtained from the tests included the nanoparticle size, potential and the remaining content of Cur in the nanoparticles. The initial content of Cur was set at 100%, so it was not reflected in the data presented in this study.

### 2.5. Redispersibility 

The freeze-dried samples (Z-cur, Z/S-cur, Z/S/C-cur nanocarriers) were redispersed in ultrapure water and dissolved using sonication until fully dissolved. The particle size, PDI, zeta potential and drug loading efficiency (EE%) were then measured and compared with the freshly prepared samples to evaluate the redispersibility of Z-cur, Z/S-cur and Z/S/C-cur nanocarriers.

### 2.6. Simulated Gastrointestinal Release

The experimental protocol for in vitro gastrointestinal digestion was modified based on the previous work [41]. To prepare simulated gastric fluid (SGF), 3.2 mg/mL pepsin and 2 mg/mL NaCl were adjusted to pH 2 with hydrochloric acid. To prepare simulated intestinal fluid (SIF), 12 mg/mL bile salt, 2 mg/mL pancreatin, and 8.8 mg/mL NaCl were mixed and adjusted to pH 7.5 with hydrochloric acid and sodium hydroxide. To perform the experiment, 2 mL of each group of nanoparticles loaded with Cur was added to 20 mL of SGF and digested at 37 °C for 90 min, followed by its addition to SIF and further digestion for 270 min with sampling every 30 min for analysis (particle size analysis was performed as described in Section 2.3.1). Cur release was measured by centrifuging the collected samples at 9800 rpm to remove insoluble materials, and the remaining Cur in the supernatant was determined using UV analysis.

### 2.7. MTT

To evaluate the biosecurity of Z/S/C nanoparticles, it is necessary to examine their cytotoxicity. In this study, the MTT assay was carried out using L-O2 and HepG2 cells. Briefly, L-O2 and HepG2 cells were seeded in 96-well plates and incubated until the cell number reached around 70%. Various concentrations of the nanoparticles (200, 400, 800 and 1200 μg/mL) were added and incubated for 24 h at 37 °C. MTT (5 mg/mL) was then added and incubated for an additional 4 h (control group was also included). Finally, DMSO was added to stop the incubation, and the absorbance of each group was measured using a microplate reader at 490 nm to determine cell viability.

### 2.8. Statistical Analysis 

All the data were measured at least 3 times, and the provided data were shown as the mean value ± standard deviation (SD). The significant difference was calculated using the one-way ANOVA test through Prism 7.0 software. The *p* value was less than 0.05.

## 3. Result and Discussion

### 3.1. Preparation of the Zein-Based Nanoparticles

#### 3.1.1. Effect of Mass Ratio of Zein and SPI on Nanoparticles 

At the pre-experiment stage, different concentrations of zein (5 mg/mL, 10 mg/mL, 15 mg/mL and 20 mg/mL) were tested to prepare the zein nanoparticles in Appendix A. A total of 10 mg/mL was selected as the optimal concentration for the final experiment, as it resulted in the smallest particle size and lower PDI value compared to other concentrations. Next, Z/S binary nanoparticles were constructed using zein and SPI at different ratios, i.e., 4:1, 2:1, 1:1, 2:3, 1:2 and 2:5. The nanoparticles were prepared using a pH-driven method, and the average size and zeta potential were measured at pH 7.5. As shown in Figure 1A, the average size of the zein nanoparticles is 152 nm with a zeta potential of −49.93 mV, which is consistent with previous research findings [42]. This confirmed that the zein nanoparticles were successfully constructed by the pH-driven method.

The results of zeta potential experiments showed that the zeta potential of 3 mg/mL SPI in neutral aqueous solution was −39.3 mV. In the case of Z/S binary nanoparticles, as the mass ratio of SPI increased, the zeta potential (absolute value) of Z/S composite nanoparticles did not show significant changes at first, but it continuously decreased with the concentration of SPI. Meanwhile, it decreased in the end and finally remained constant at −39.23 mV in Figure 1B. The phenomenon suggested that SPI gradually wrapped zein nanoparticles externally through electrostatic and hydrogen bond interactions with zein nanoparticles.

In the Z/S binary nanoparticles with a mass ratio of 4:1, 2:1 and 1:1, the zeta potential decreases continuously, but its absolute value remains higher than that of SPI in a neutral aqueous solution, as shown in Figure 1B. This indicates that isolated soy protein does not fully coat the surface of zein nanoparticles, and some zein groups remain exposed. As the concentration ratio of Z/S changed to 2:3, the zeta potential of the Z/S binary nanoparticles decreased to −39.23 mV. Further increase in SPI concentration did not cause significant change (*p* < 0.05) in the zeta potential for Z/S ratios of 1:2 or 2:5, indicating the complete encapsulation of zein nanoparticles by SPI and the formation of stable nanoparticles at the Z/S ratio of 2:3.

This finding demonstrates that SPI can co-assemble with zein nanoparticles through electrostatic and steric hindrance. However, in the assembly process, the higher hydrophobicity of zein and the more amphipathic nature of SPI cause the SPI location in the nanoparticle to be divided into two parts. The hydrophilic part of SPI interacts with zein through electrostatic interactions located outside the formed Z/S binary nanoparticles. The incorporation of the hydrophobic part of SPI into the zein nanoparticles resulted in a tighter binding between the two components [43]. As the concentration of SPI increases, the electrostatic interaction between SPI and zein became stronger, and this made the structure more stable.

The effect of SPI concentration on the particle size of Z/S binary nanoparticles is illustrated in Figure 1A. When the mass ratio of zein to SPI were 4:1 and 2:1, the sizes of Z/S binary nanoparticles increased to 157.8 nm and 181.7 nm, respectively, indicating that the lower part of SPI is deposited on the surface of zein nanoparticles, whereas the particle size only increases without any significant difference due to the weak electrostatic interaction between zein and SPI. Nevertheless, with the Z/S ratio increased to Z/S 1:1, the particle size of Z/S nanoparticles decreased to 105.3 nm, and further decreased to a minimum of 84.7 nm at a 2:3 ratio of Z/S. At higher SPI concentrations, the particle size of Z/S 1:2 was 92.84 nm, and the particle size of Z/S 2:5 was 109.6 nm.

When the concentration of SPI exceeds a certain level, the surplus protein gradually accumulates on the exterior of Z/S nanoparticles, resulting in the particle size increasing. These findings indicated that SPI can interact with the surface of zein nanoparticles through electrostatic and hydrogen bond interactions at a specific concentration, leading to the formation of a dense nanoparticle structure. Appendix A provided a visualization of the particle size, PDI and peak morphology of Z/S nanoparticles with various mass ratios. As noted in the literature [44], a smaller PDI indicated better and more stable nanoparticle dispersion. Conversely, a PDI > 0.3 implied an increased likelihood of nanoparticle agglomeration and poor dispersion. Based on PDI analysis, Z/S 2:3 nanoparticles demonstrated the smallest particle size and lower PDI value. Considering both the particle size and zeta potential results, Z/S 2:3 nanoparticles were chosen for further investigation.

#### 3.1.2. Effect of Mass Ratio of SPI and Sodium CMC-Na on Nanoparticles

Z/S/C ternary nanoparticles were prepared using different mass ratios of SPI and CMC-Na (4:1, 2:1, 1:1, 1:2, 2:3, 2:5). The particle size, PDI and zeta potential were measured. CMC-Na is a branched polysaccharide that is a derivative of cellulose with some of its hydroxyl groups substituted by -CH_2_COOH groups, with a pKa value of approximately 3. As a polyanionic polymer, CMC-Na has a strong negative charge in neutral aqueous solutions. When tested, the zeta potential of a 3 mg/mL neutral aqueous solution of sodium carboxymethylcellulose was found to be −62.8 mV. However, the absolute value of the zeta potential of the resulting Z/S/C nanoparticles was lower than that of the CMC-Na solution. This is likely due to the electrostatic interaction between CMC-Na and local positively charged residues on zein. The combination of opposite charges resulted in the absolute value of the zeta potential of the Z/S/C ternary nanoparticles decreasing [45].

As the mass ratio of SPI/CMC-Na changed, the concentration of CMC-Na gradually increased, resulting in particle size changes as follows: 4:1 (79.65 nm), 2:1 (73.9 nm) and 1:1 (71.41 nm) nanoparticles. As shown in Figure 2A, the particle size of Z/S/C ternary nanoparticles was smaller than that of Z/S binary nanoparticles at this mass ratio, with the smallest mass ratio of particle size being SPI/CMC-Na as 1:1. However, the particle size also increased as the ratio gradually shifted towards CMC-Na. When the mass ratios of SPI to CMC-Na were 1:2 (89.78 nm), 2:3 (102.2 nm) and 2:5 (113.3 nm), the particle diameters of Z/S/C ternary nanoparticles were larger than those of Z/S binary nanoparticles.

Excessive deposition of CMC-Na on the surface of nanoparticles may lead to an increase in particle size. Furthermore, although SPI/CMC-Na resulted in smaller particle sizes than Z/S binary nanoparticles at 4:1 and 2:1 with lower LE% values, the PDI of Z/S/C ternary nanoparticles at this mass ratio was higher than 0.3. This suggested that the Z/S/C ternary nanoparticles at this ratio are not evenly distributed and that the CMC-Na is not entirely covering the exterior of the Z/S binary nanoparticles, which lead to an unstable formation of the ternary nanoparticles.

The Z/S/C ternary nanoparticles increased continuously in the absolute value of the zeta potential, as depicted in Figure 2B. The zeta potential value approached equilibrium with the CMC-Na concentration, reaching a steady state with the mass ratio of SPI to CMC-Na at a 1:1 ratio. Based on these results, the Z/S/C ternary nanoparticles prepared at the 1:1 mass ratio of SPI/CMC-Na have a more uniform distribution compared to those with other mass ratios. The PDI value of the Z/S/C ternary nanoparticles is less than 0.3, with a smaller particle size of 71.41 nm and a more uniform surface charge of −49.97 mV. Thus, to select nanocarriers that are smaller in size and more stable, the optimal mass ratio of Z/S/C was chosen as 2:3:3 for loading turmeric in the zein ternary nanoparticles. The Appendix A depicts the particle size distribution of Z/S/C ternary nanoparticles at various SPI/CMC-Na mass ratios.

### 3.2. Characterization of Complex Nanoparticles

#### 3.2.1. Transmission Electron Microscope (TEM)

The electron microscope images of Z, Z/S and Z/S/C nanoparticles are depicted in Figure 3 at 100× and 200× magnification. All nanoparticle structures exhibit spherical shapes. It is noticeable that pure zein nanoparticles tend to aggregate and form clusters, mainly due to hydrophobicity between zein nanoparticles under neutral conditions. Moreover, zein is primarily composed of nonpolar amino acids, which results in the formation of chain structures of zein under electron microscopy [46]. From the electron microscope image of Z/S nanoparticles, it is evident that SPI acts as a stabilizer to coat the surface of the zein nanoparticles and some excess stabilizer is dispersed outside the nanoparticles and accumulated on their surface. When CMC-Na is used to form Z/S/C ternary composite nanoparticles, as shown in Figure 3, the surface of the resulting nanoparticles appears smooth, and their size is smaller than that of Z/S nanoparticles. Furthermore, the excessive SPI stabilizer is enclosed within the nanoparticles, making them more stable, evenly distributed and well-dispersed.

This uniform spherical shape may promote cell uptake after administration and the efficiency of drug administration to cells [47]. It can also be observed that the center of Z/S/C nanoparticles is darker than that of Z/S nanoparticles under microscopic imaging of the nanoparticles, indicating that the formed Z/S/C nanoparticles have a tighter structure and a higher thickness.

#### 3.2.2. Particle Size, Zeta Potential and Encapsulation Efficiency

The particle size, polydispersity index (PDI), zeta potential and EE% of Z-cur, Z/S-cur and Z/S/C-cur nanoparticles are summarized in Table 2. It can be observed that the particle size of Cur-loaded nanoparticles is smaller than that of nanoparticles without drug loading, such as Z, Z/S and Z/S/C. This is because Cur, when deprotonated in an alkaline solution, becomes negatively charged and interacts with zein nanoparticles, which have a certain negative charge on their surface. Cur is encapsulated inside the zein nanoparticles, making the formed nanoparticles more compact through electrostatic and steric effects. This is consistent with previous studies [48]. 

Once prepared, the EE% of zein nanoparticles was found to be only 50.80%, which may be due to being near the isoelectric point (pI) of zein 6.2, causing the nanoparticles to aggregate and Cur to leak out of the nanoparticles. The addition of SPI and CMC-Na changes the pI of the complex nanoparticles and increases the EE%. The EE% of Z/S-cur and Z/S/C-cur nanoparticles were found to be 82.20% and 90.90%, respectively. The higher EE% of Z/S/C-cur ternary nanoparticles compared to Z/S-cur binary nanoparticles may be due to the water-soluble polysaccharide CMC-Na, which can capture some Cur in the relatively hydrophilic regions of the nanoparticles during the pH-driven method used to load Cur into the nanoparticles. This is consistent with previous research reports [28]. The PDI values of all formed nanoparticles were less than 0.3, indicating that all formed nanoparticles were uniformly distributed. Additionally, the high zeta potential values (>30 mV) of the nanoparticles indicate that they have sufficient electrostatic repulsion, which can prevent them from aggregation with each other, thus ensuring the stability of the nanoparticles [49]. The changes in particle size and PDI of Z-cur, Z/S-cur and Z/S/C-cur nanoparticles after drug loading are shown in the Appendix A, and their room temperature status can be seen in Figure 4.

#### 3.2.3. FTIR

The individual components of zein, SPI, CMC-Na and Cur bioactive substances were analyzed using Fourier-transform infrared spectroscopy (FTIR), as well as the driving forces behind the interaction of these bioactive substances in the formation of nanocomposites. As shown in Figure 5A,B, all samples exhibited a broad absorption peak in the range of 3100–3500 (cm^−1^), which represents the stretching and vibration of O−H bonds, consistent with previous studies [50]. In Figure 5A, we observed absorption peaks of the O-H group of CMC-Na, SPI and zein at 3360, 3270 and 3288 cm^−1^, respectively. With the formation of the nanocomposite particles Z-cur, Z/S-cur and Z/S/C-cur, the position of the O−H bond in the complex nanocarrier shifted, gradually moving to 3288, 3291 and 3280 cm^−1^. This suggests that there are hydrogen-bonding interactions among these four substances, which is consistent with previous studies [42]. The characteristic peaks of zein in the spectrum shown in Figure 5A are 1644 (cm^−1^) and 1530 (cm^−1^), representing amide I (C=O stretching) and amide II (N-H interaction), respectively. After the addition of Cur and SPI, the amide I peak of Z/S-cur shifted to 1639 (cm^−1^), and the amide II peak shifted to 1536 and 1515 (cm^−1^), indicating the existence of electrostatic interactions among zein, SPI, and Cur. At the same time the change of amide bond indicates that the addition of curcumin changes the configuration of the protein. Upon the formation of Z/S/C-cur nanoparticles after adding CMC-Na, the characteristic peaks of CMC-Na at 1589, 1410 and 1324 cm^−1^ shifted to 1590, 1414 and 1325 cm^−1^, respectively. This suggests that the outermost layer of CMC-Na interacts electrostatically with the intermediate layer of SPI, causing peak shifts. The changes in amide bonds have also been reflected in previous studies [51]. When zein and rhamnolipid interact, a shift in amide II band can be observed. In Figure 5A, a characteristic peak for the hydroxyl group vibration of phenolic of Cur at 3504 cm^−1^ appears. In the spectra of Figure 5B, Cur shows its characteristic peaks at 3504, 1599, 1509, 1427, 1154 and 812 cm^−1^, which is consistent with previous studies [52]. As the three types of complex nanoparticles were formed, the characteristic peak of Cur gradually disappeared within each complex nanoparticle and only the characteristic peaks of each nanoparticle were observed. This indicates that Cur was encapsulated in the hydrophobic shell of the zein nanoparticles by hydrophobic interactions [33]. This result is consistent with previous study, where after the formation of zein–casein–fucoidan composite nanocarriers, the characteristic peak of Cur disappeared in the nanocarrier.

#### 3.2.4. XRD

This article presents the XRD information of Cur, CMC-Na, SPI, zein and three formed composite nanoparticles. XRD mainly examines the diffraction and crystallinity information of the samples. As shown in Figure 6 and Appendix A, zein and SPI, as protein samples, exhibited two broad peaks near diffraction angles of 9° and 20°, indicating the presence of the amorphous form of protein samples. CMC-Na only has a flat peak near 20.2°, indicating the amorphous form of natural polysaccharides, which is consistent with previous studies [53]. Cur exhibits characteristic peaks at 8.9°, 12.2°, 14.5°, 17.3°, 23.4°, 24.5° and 25.5°, indicating the existence of a crystal structure [30]. With the formation of nanoparticles, the characteristic peaks of Cur in Z-cur, Z/S-cur, and Z/S/C-cur nanoparticles disappear, indicating the interaction of Cur with the three nanoparticles through hydrophobic or hydrogen bonding, and the result suggests that Cur is encapsulated in the three nanoparticles in an amorphous form. Meanwhile, it was found that the characteristic peaks of zein, SPI and CMC-Na weaken or disappear after the formation of Z-cur, Z/S-cur and Z/S/C-cur nanoparticles, indicating the molecular interactions between Cur, zein, SPI and CMC-Na. This is consistent with previous studies [54], in which the characteristic peaks of sodium caseinate and alginates disappear after using alginates to encapsulate zein–sodium caseinate nanoparticles. 

### 3.3. Stability Research

#### 3.3.1. pH Stability

As the application of nanoparticles will expose them to different environments, such as varying pH levels and potential contamination from extraneous substances, the stability of the nanoparticles can be affected. For instance, in coastal cities, the high humidity levels in the atmosphere can contain ions from mineral water vapor, which can also impact the stability of the nanoparticles. Therefore, to ensure the stability of nanoparticles during transport, delivery and commercialization, it is crucial to consider the impact of pH and ionic strength.

Based on the pH stability of the different types of nanoparticles, Z-cur, Z/S-cur and Z/S/C-cur, as shown in Figure 7A–C, it can be seen from Figure 7A that Z-cur nanoparticles undergo flocculation at pH 6 and pH 7, resulting in particle sizes exceeding the nanometer range that are difficult to measure with instruments. At pH 6, the potential was 3.6 mV, which is consistent with the theoretical isoelectric point (pI) of zein (6.2). The low surface charge and precipitation near the protein isoelectric point led to an increase in particle size and turbidity, which is consistent with previous studies [55]. Due to the decrease in zeta potential, the electrostatic and steric hindrance between nanoparticles decreased, resulting in extremely unstable nanoparticles and aggregation phenomena. Similarly, for Z/S-cur nanoparticles, an increase in particle size and precipitation were also observed at pH 4 and pH 5, which can be explained by the pI of the SPI wrapped around the Z/S-cur (4.5) [56]. At the pH where Z-cur nanoparticles precipitated, Z/S-cur nanoparticles did not show any significant changes in particle size, further indicating that SPI was wrapped around the outside of zein to form Z/S-cur binary nanoparticles.

Compared to Z-cur and Z/S-cur nanoparticles, Z/S/C-cur nanoparticles exhibited a wide range of pH stability. Due to the pKa value of CMC-Na being around 3 [57], the turbidity of Z/S/C-cur nanoparticles continuously increased as shown in Figure 7C near pH 3, and even precipitated at pH 2. The reason for this precipitation is that the surface charge of Z/S/C-cur ternary nanoparticles is low and cannot form strong electrostatic repulsion, leading to nanoparticle aggregation. However, at pH 3, we found that although the particle size and turbidity of Z/S/C-cur nanoparticles increased, the particle size remained in the nanoscale range at 153.9 nm, indicating that Z/S/C-cur ternary nanoparticles exhibited good stability at pH 3–9. This also suggests that the addition of CMC-Na improved the stability of Z/S-cur binary nanoparticles. Furthermore, the addition of SPI and CMC-Na has a synergistic effect on the formation of zein nanoparticles.

#### 3.3.2. Ionic Strength Stability

In comparison between Z/S-cur and Z/S/C-cur nanoparticles, their particle size and appearance were tested at concentrations ranging from 0 to 105 mmol/L, as shown in Figure 8A. Subsequently, the particle size and appearance of Z/S-cur and Z/S/C-cur nanoparticles were tested at concentrations ranging from 0 to 250 mmol/L, as shown in Figure 8B. The results showed that the particle size of Z-cur nanoparticles began to increase significantly when the concentration of NaCl rose to 84 mmol/L. In contrast to a study by [58], pH cycling was found to have a stronger tolerance for ions than the anti-solvent method used to prepare zein nanoparticles. The study showed that [52] nanoparticles prepared using the anti-solvent method tended to aggregate when the NaCl concentration exceeded 20 mmol/L. This indicates the advantage of pH cycling over the anti-solvent method. Indicated under the concentration range of 0–105 mmol/L, the particle size and appearance of Z/S-cur nanoparticles remained stable.

When comparing Z/S-cur and Z/S/C-cur nanoparticles, it was found that at concentrations ranging from 0 to 250 mmol/L, the particle size of Z/S-cur nanoparticles began to change slightly, increasing by 101.8 nm (*p* > 0.05), indicating that the presence of Na^+^ ions in NaCl weakened the electrostatic attraction between zein, SPI and CMC-Na [55]. When the concentration continued to rise to 250 mmol/L, the nanoparticles turbidity and particle size changed significantly. From the appearance, it was found that Z/S-cur nanoparticles showed sedimentation, indicating that the range of ion tolerance for Z/S-cur was 0–200 mmol/L. In contrast, Z/S/C-cur nanoparticles showed only a slight increase in particle size at 200 mmol/L and 250 mmol/L, with no significant difference from normal Z/S/C-cur ternary nanoparticles. These results suggest that nanoparticles stabilized by CMC-Na and SPI exhibit excellent ion strength stability, and Z/S/C-cur ternary nanoparticles have a broader ion strength tolerance than Z/S-cur binary nanoparticles.

#### 3.3.3. Photochemical Stability

As a class of substances highly sensitive to ultraviolet (UV) radiation, Cur rapidly degrades when exposed to UV light for an extended period of time. A study reported that the Cur content drops to below 5% when exposed to UV light for over 30 hours [59]. As a novel plant protein carrier, zein contains double bonds and aromatic amino acid residues that can enhance the nanoparticle’s ability to absorb UV light [40]. Meanwhile, the hydrogen bonds formed directly between Cur and CMC-Na and SPI can also increase its UV absorption and improve its stability. As shown in Figure 9D, pure Cur decreased to 5.73% after 180 minutes of UV irradiation, while Z-cur demonstrated stronger resistance to UV irradiation with a retention rate of 29.17%. It is worth noting that both Z/S-cur and Z/S/C-cur nanoparticles exhibited high Cur retention rates of 60.53% and 75.13%, respectively, after UV irradiation. It suggested that adding SPI and CMC-Na can increase the protective effect on nanoparticles and slow down the degradation of Cur under UV light. The interaction between SPI, CMC-Na and the surface of zein nanoparticles increases the electrostatic repulsion and steric hindrance of the nanoparticles, forming a dense structure on the surface of zein nanoparticles, making them more stable. However, compared to Z/S-cur binary nanoparticles, Z/S/C-cur nanoparticles have a more compact and dense structure, resulting in a stronger ability to resist UV absorption.

#### 3.3.4. Storage Stability

We examined the stability of the three types of nanoparticles, Z-cur, Z/S-cur and Z/S/C-cur, to assess the duration of storage and transportation of nanoparticles, as well as their potential commercial value. The particle size, zeta potential and retention of Cur in the three types of nanoparticles were measured using the methods established previously after storage at 4 °C for 7 and 14 days, respectively.

There were no significant changes in particle size of the Z-cur nanoparticles being discovered during the first 7 days of storage, with the size increasing from 144.5 nm to 151.1 nm (*p* > 0.05), but a significant increase was observed after 14 days of storage, with the size increasing to 193.7 nm (Figure 9A,B). Additionally, a phenomenon was observed where the absolute value of the zeta potential of Z-cur gradually decreased from 48.83 mV to 34.67 mV as the storage time increased. This is likely due to the instability of the nanoparticles, resulting in poor dispersion homogeneity and a decrease in surface potential and electrostatic binding ability [60].

Furthermore, sedimentation was observed at the bottom of the container after 14 days of storage. However, there was not much difference in the particle size and zeta potential of Z/S-cur and Z/S/C-cur nanoparticles throughout the storage period (*p* < 0.05), indicating good storage stability of the nanoparticles stabilized by SPI and CMC-Na. As shown in Figure 9C, the retention rate of free Cur after 14 days of storage was 34%. However, after encapsulation within nanoparticles, the retention rates of Cur in Z-cur, Z/S-cur and Z/S/C-cur nanoparticles increased to 61.50%, 74.23% and 80.83%, respectively. This indicates that the stability of free Cur was enhanced after encapsulation within nanoparticles. Nonetheless, from the retention rate changes of Cur in Z/S-cur and Z/S/C-cur after 7 and 14 days, it was observed that the retention rate of Cur in Z/S-cur decreased from 87.17% to 74.23% after 14 days, while that of Z/S/C-cur decreased from 90.53% to 80.83% after 14 days. The results also showed that the drug leakage rate of Cur from Z/S/C-cur nanoparticles was slower than that of Z/S-cur nanoparticles, indicating that the electrostatic interactions of Z/S/C-cur were stronger and that the nanoparticles were more stable. We can infer that adding SPI and CMC-Na improves the physical stability of the ternary nanoparticles through electrostatic interactions and steric hindrance. Moreover, as a type of hydrophilic polysaccharide, its combination with CMC-Na can improve the hydrophilicity of zein nanoparticles, prevent aggregation under hydrophobic conditions and capture Cur in a relatively hydrophilic environment, thereby preventing Cur from leaking out of the nanoparticles.

### 3.4. Redispersibility

Due to the hydrophobic structure of zein under neutral conditions, which is caused by the unique nonpolar amino acids in it, aggregation often occurs during the freeze-drying and transportation process, presenting a challenge to the product’s transportation and commercialization [61]. To address this phenomenon, protein cryoprotectants, such as sucrose [62], are often applied to prevent the aggregation of zein nanoparticles. In this study, the Z-cur, Z/S-cur and Z/S/C-cur nanoparticles were evaluated for their nanosize, PDI, zeta potential and drug loading (EE%) before and after freeze-drying and redispersion. As shown in Table 3, the nanosize of the Z-cur nanoparticles changed significantly from 144.4 nm to 232.4 nm, and the PDI increased from 0.19 to 0.46 after freeze-drying, which indicated that the nanodispersion system became extremely unstable. 

The extensive leakage of Cur also demonstrated that the Z-cur nanoparticles alone could not withstand the changes caused by freeze-drying. However, with the addition of SPI and CMC-Na, the difference in the change of Z/S-cur and Z/S/C-cur nanoparticles before and after freeze-drying was smaller, indicating that the newly formed nanoparticles could resist the aggregation of nanoparticles caused by freeze-drying and demonstrating that SPI and CMC-Na act as cryoprotectants. Compared to the Z/S-cur nanoparticles, the Z/S/C-cur nanoparticles exhibited smaller differences in particle size and drug loading, indicating that they were more stable. The addition of stabilizers did not significantly affect the drug loading of the zein nanoparticles, which was consistent with the findings of a previous study [61] that found that the drug loading of zein nanoparticles stabilized by sodium caseinate was not significantly affected after redispersion following freeze-drying.

### 3.5. Simulating Gastrointestinal Release

In order to systematically evaluate the release of Cur in the human gastrointestinal tract, we investigated the nanoparticle size of Z-cur, Z/S-cur and Z/S/C-cur nanoparticles at different stages of digestion in the gastrointestinal tract, as well as the release curves of the Cur control group and each group of Cur-loaded nanoparticles. As shown in Figure 10A, after 90 min of digestion, the three types of nanoparticles, Z-cur, Z/S-cur and Z/S/C-cur, exhibited a significant increase in size, but remained at the level of nanoparticles (<1 μm). However, there was a noticeable change in nanoparticle size compared with before digestion, possibly due to digestion by gastric proteases. When digestion began, the nanoparticles were hydrolyzed by gastric proteases, breaking down noncovalent bonds into small peptide molecules. The size of Z-cur nanoparticles increased significantly as the drug transitioned from SGF to SIF after 90 min of digestion, but with prolonged digestion time, the size showed a certain downward trend. Z/S-cur and Z/S/C-cur nanoparticles also exhibited similar trends. The reason for this change may be due to the rapid disintegration of nanoparticles in simulated intestinal fluid by pancreatic enzymes caused by the drastic pH change as the nanoparticles transitioned from simulated gastric fluid to simulated intestinal fluid. It resulted in a large amount of charge rushing into the nanoparticles, which destroyed the structure of the nanoparticles and lowered the surface charge, causing nanoparticle aggregation and a significant increase in size. However, as pancreatic enzymes hydrolyzed, some hydrolyzed products, such as proteins and peptides, reformed into nanoparticle micelles and new complexes with bile salts. Herein, the size of the nanoparticles decreased with prolonged digestion time [63,64,65].

The process of gastrointestinal digestion is extremely complex, with different release times and locations having significant impacts on the utilization of bioactivity. Cur, as a hydrophobic bioactive substance, is generally only absorbed and utilized by the human body in the intestine. Therefore, it is necessary to avoid the premature release of Cur in simulated gastric fluid, which reduces the bioavailability of the drug. As shown in Figure 10B, 52.92% of free Cur was released into SGF conditions within 90 minutes, while Z-cur, Z/S-cur and Z/S/C-cur nanoparticles released 28.54%, 17.37%, and 14.11%, respectively. Subsequently, under SIF conditions, digestion continued until 270 minutes, with free Cur released up to 92.67%. Z-cur nanoparticles released 72.45%, which is consistent with previous research [51]. The release of Cur from Z/S-cur and Z/S/C-cur nanoparticles was 45.16% and 39.03%, respectively. Pure Cur is rapidly released due to physical and chemical driving forces, but encapsulating Cur in nanoparticles provides a certain barrier effect, significantly reducing the release rate. Research on zein nanoparticles that encapsulate Cur [66] has shown that these nanoparticles have a higher solubility in SIF, indicating a faster Cur release rate compared to Z/S-cur and Z/S/C-cur in simulated intestinal fluid. As shown in Figure 10B, with the incorporation of SPI and CMC-Na, the release efficiency of Cur gradually decreased compared to pure Z-Cur, and the release time was extended. It suggested that the excessive crosslinking between SPI, CMC-Na and zein nanoparticles were captured deeper inside the nanoparticles, which made the release slowed down, and thereby increased the bioavailability of Cur.

### 3.6. Cytotoxicity MTT

To assess the impact and toxicity of Z/S/C ternary nanoparticles on the human body, HepG2 and L-O2 cells were selected for evaluation and testing at different concentrations, as shown in Figure 11. After incubating the cells for 24 h with drug concentrations of 200, 400, 800 and 1200 ug/mL, the remaining drugs on the 96-well plate were washed with PBS and MTT was added for toxicity testing. The results showed that the cell viability at all concentrations was above 93%, indicating that Z/S/C nanoparticles have some biocompatibility in the human body [28]. The pH-driven method used to construct Z/S/C ternary nanoparticles is safe for in vivo administration, and its advantage of not containing organic solvents compared to the antisolvent method is also one of the judgment criteria for its enormous potential in the pharmaceutical and food industries.

## 4. Conclusions

In summary, the Z/S/C NPs ternary composite nanoparticles were prepared using an improved pH-driven method. After preparation, Cur was encapsulated and delivered as a bioactive substance. CMC-Na provided a relatively hydrophilic environment, allowing Cur to be trapped in the relatively hydrophilic outer shell, increasing the drug loading of Cur. Additionally, the introduction of SPI and CMC-Na not only improved the solubility of Cur but also increased the interaction of hydrogen bonds and electrostatic forces with the zein nanoparticle surface, improving its physicochemical stability. After comparing the three types of nanoparticles, Z/S/C-cur showed superior protective effects for Cur in terms of photo-stability, storage stability, and simulated gastrointestinal digestion. Z/S/C-cur also showed a significant improvement in tolerance to ion strength compared to Z/S-cur and Z-cur. The biocompatibility of Z/S/C-cur with human cells demonstrated the potential application of the nanoparticles. This study provides valuable information for the encapsulation of hydrophobic compounds using the pH cycling method to construct ternary composite nanoparticles and have potential use in the food industry in the future.

## Figures and Tables

**Figure 1 foods-12-02692-f001:**
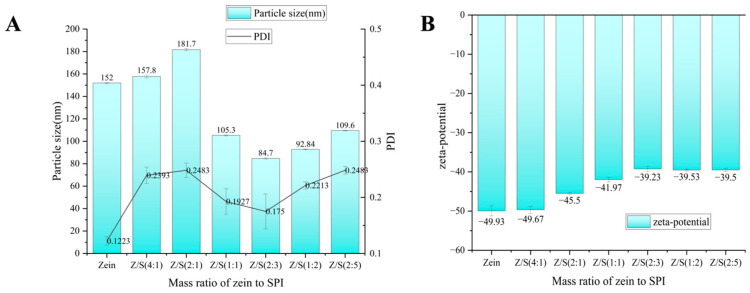
Z/S nanoparticles formed by zein and different proportions of SPI. (**A**) Mass ratio of zein to SPI particle size. (**B**) Zeta-potential of Z/S nanoparticles. Data represent mean of three experiments.

**Figure 2 foods-12-02692-f002:**
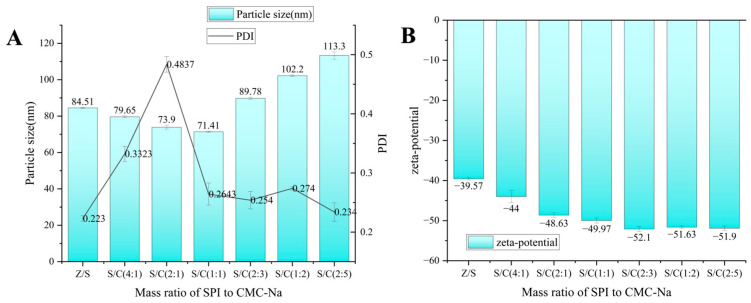
(**A**) Different mass ratio of SPI to CMC-Na on formed Z/S/C nanoparticle particle size and PDI. (**B**) Zeta-potential of different mass ratio of SPI to CMC-Na. Data represent mean of three experiments.

**Figure 3 foods-12-02692-f003:**
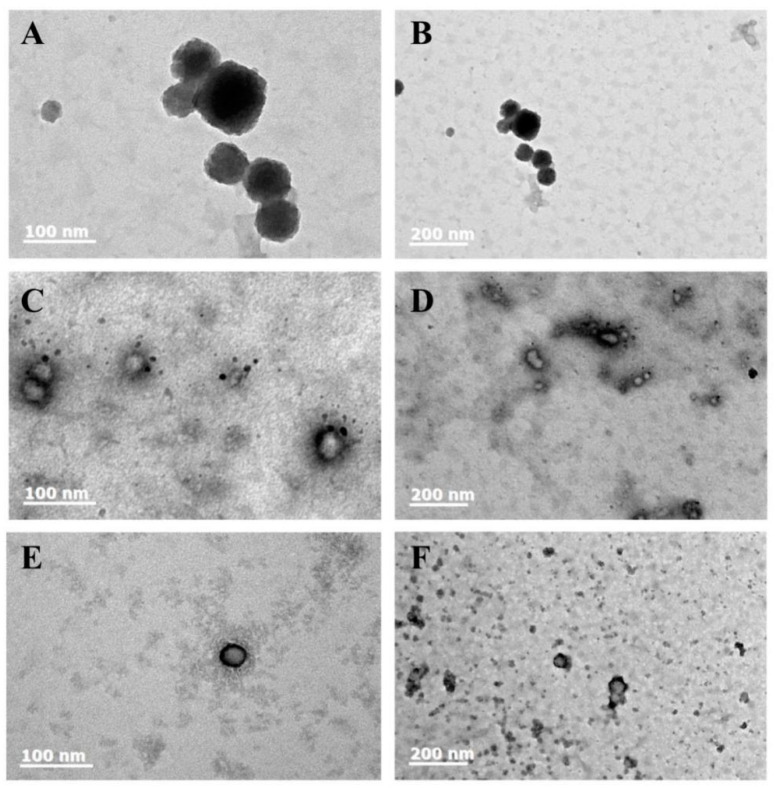
(**A**,**B**) Appearance of zein nanoparticles (without phosphotungstic acid) under 200× and 100× electron microscope; (**C**,**D**) appearance of Z/S nanoparticles; (**E**,**F**) Z/S/C nanoparticles formed at 200× and 100× (containing 2% phosphotungstic acid).

**Figure 4 foods-12-02692-f004:**
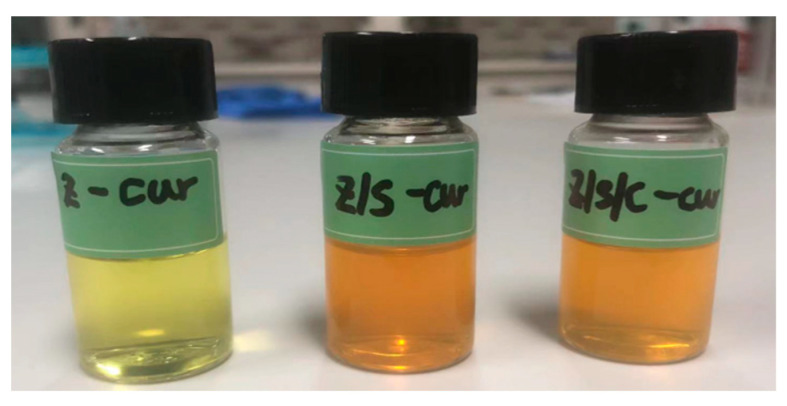
The appearance of the different group nanoparticles at room temperature after the Z-cur, Z/S-cur and Z/S/C-cur nanoparticles have been prepared.

**Figure 5 foods-12-02692-f005:**
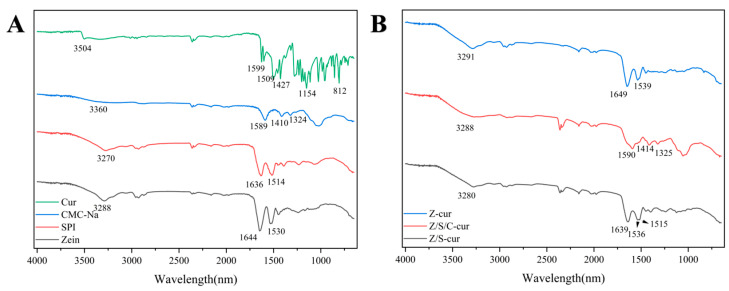
(**A**) FTIR of Cur, CMC-Na, SPI and zein; (**B**) the spectra of Z-cur, Z/S/C-cur and Z/S-cur nanoparticles.

**Figure 6 foods-12-02692-f006:**
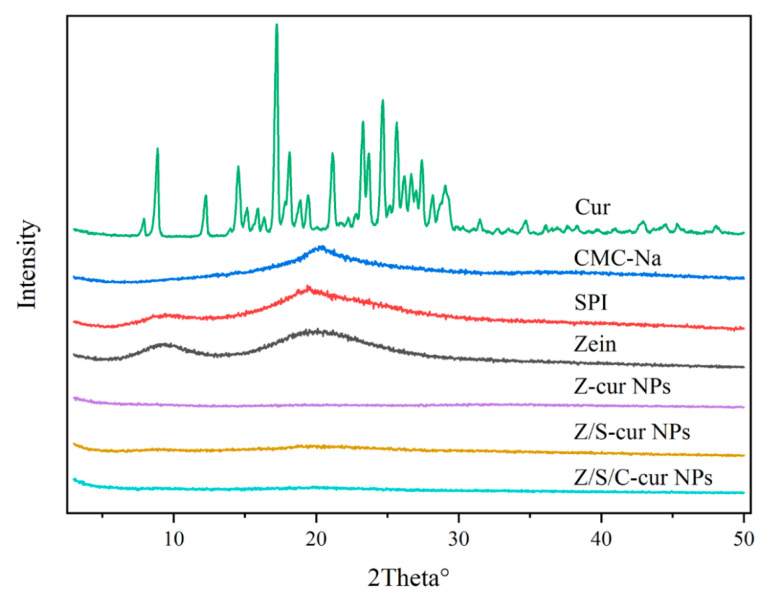
XRD spectra of three nanoparticles, Z-cur, Z/S-cur and Z/S/C-cur, and XRD spectra of four bioactive components, Cur, CMC-Na, SPI and zein.

**Figure 7 foods-12-02692-f007:**
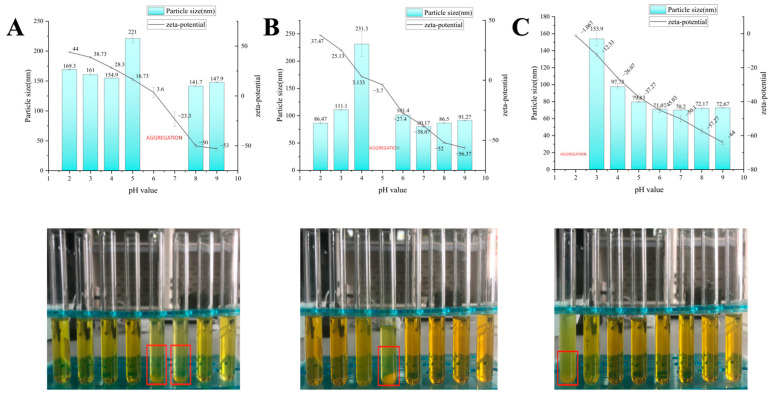
(**A**) Z-cur nanoparticles; (**B**) Z/S-cur nanoparticles; (**C**) Z/S/C-cur nanoparticles. Change and appearance of the particle size and zeta potential of the three nanoparticles under pH 2–9. We can see the sediment at the bottom of the glass bottle use red box.

**Figure 8 foods-12-02692-f008:**
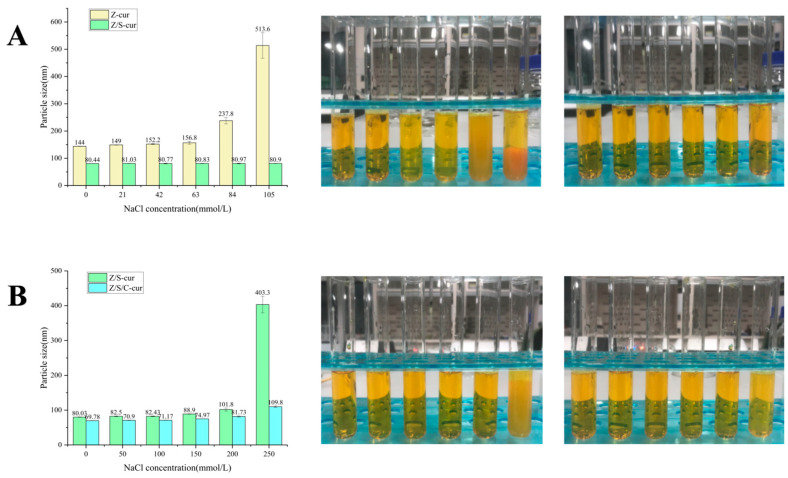
(**A**) Comparison between the particle size and appearance of Z-cur nanoparticles and Z/S-cur nanoparticles at (0–105 mmol/L) ionic concentrations; (**B**) comparison between the particle size and appearance of Z/S-cur nanoparticles and Z/S/C-cur nanoparticles at ionic concentrations of 0–250 mmol/L.

**Figure 9 foods-12-02692-f009:**
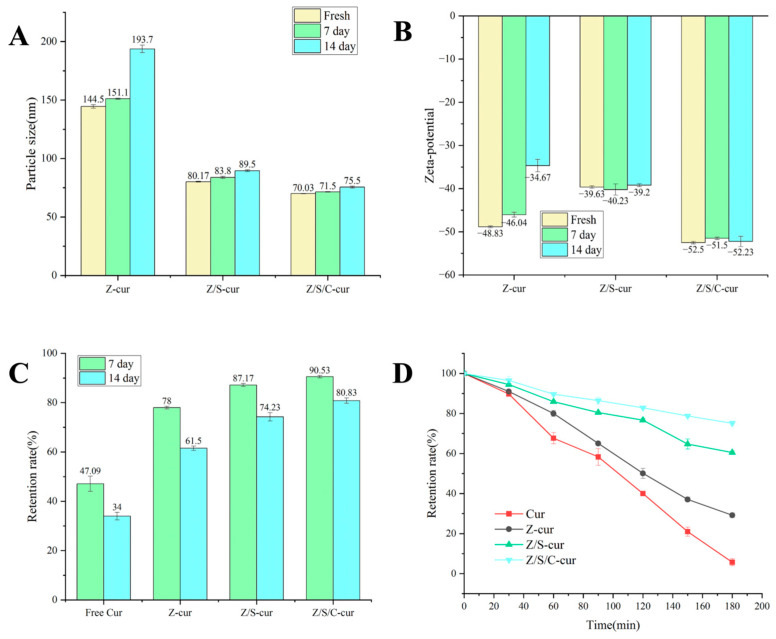
(**A**,**B**) Changes in particle size and zeta potential of three nanoparticles, Z-cur, Z/S-cur and Z/S/C-cur, stored for 7 d and 14 d at 4 °C; (**C**) retention rate of free Cur (in ethanol) and Cur of Z-cur, Z/S-cur and Z/S/C-cur nanoparticles after 7 d and 14 d; (**D**) Cur retention rate between different nanoparticles sealed with free Cur and Cur under UV irradiation.

**Figure 10 foods-12-02692-f010:**
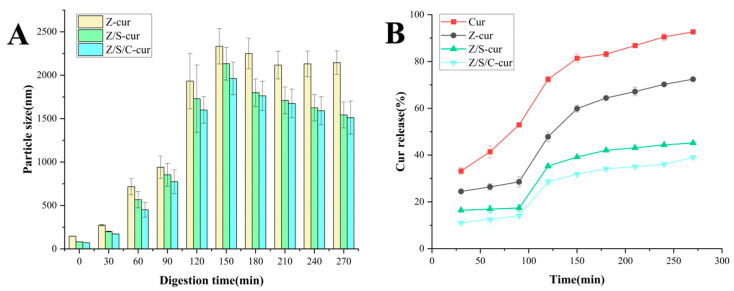
(**A**) The particle size change of Z-cur, Z/S-cur and Z/S/C-cur nanoparticles at different digestion times and stages; (**B**) the release of Cur from Z-cur, Z/S-cur and Z/S/C-cur nanoparticles at different digestion times and stages.

**Figure 11 foods-12-02692-f011:**
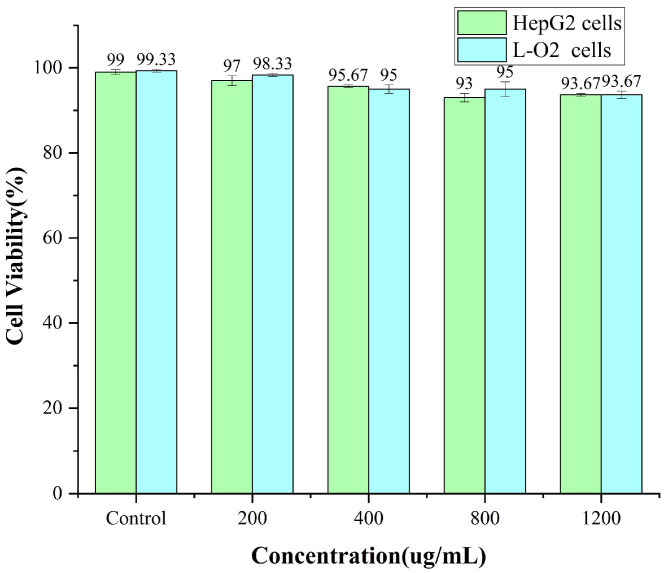
Cytotoxicity of Z/S/C nanoparticles in HepG2 cells and L-O2 cells at different concentrations.

**Table 1 foods-12-02692-t001:** Different mass ratios of the prepared Z, Z/S and Z/S/C nanoparticles.

Sample	Concentration of Zein (mg/mL)	Concentration of SPI (mg/mL)	Concentration of CMC-Na (mg/mL)	The Ratio of Zein to SPI	The Ratio of SPI to CMC-Na
zein nanoparticles	10	-	-	-	-
Zein:SPI (4:1)	10	2.5	-	4:1	-
Zein:SPI (2:1)	10	5	-	2:1	-
Zein:SPI (1:1)	10	10	-	1:1	-
Zein:SPI (2:3)	10	15	-	2:3	-
Zein:SPI (1:2)	10	20	-	1:2	-
Zein:SPI (2:5)	10	25	-	2:5	-
SPI:CMC-Na (4:1)	10	15	3.75	-	4:1
SPI:CMC-Na (2:1)	10	15	7.5	-	2:1
SPI:CMC-Na (1:1)	10	15	15	-	1:1
SPI:CMC-Na (2:3)	10	15	22.5	-	2:3
SPI:CMC-Na (1:2)	10	15	30	-	1:2
SPI:CMC-Na (2:5)	10	15	37.5	-	2:5

Note: This article uses complex nanoparticles like zein nanoparticles, zein–soy protein isolate nanoparticles, zein–soy protein isolate–carboxymethylcellulose sodium nanoparticles and therefore uses Z, Z/S and Z/S/C nanoparticles, respectively, for ease of reference. (“Z” refer to the zein;”S” refer to the soy protein isolate; “C” refer to sodium carboxymethyl cellulose).

**Table 2 foods-12-02692-t002:** The different group of nanoparticles characteristics, including particle size, PDI, Zeta-potential and EE%.

Sample	Particle (nm)	PDI	Zeta (mV)	EE (%)
Z-cur	142.80 ± 0.92	0.23 ± 0.02	−48.50 ± 0.32	50.8 ± 1.90
Z/S-cur	80.10 ± 0.58	0.24 ± 0.01	−39.50 ± 0.26	82.2 ± 0.56
Z/S/C-cur	69.80 ± 0.39	0.21 ± 0.01	−51.50 ± 0.26	90.9 ± 0.51

All data represent means ± standard deviation, and the number of sample is n = 3. At the same time, the *p* value in this table is *p* < 0.05.

**Table 3 foods-12-02692-t003:** Changes in particle size, PDI, zeta potential and EE (%) after new preparation and lyophilization of Z-cur, Z/S-cur and Z/S/C-cur.

Sample	Treatment	Particle Size (nm)	PDI	Zeta-Potential (mV)	EE (%)
Z-cur	Freshly	144.40 ± 0.91	0.19 ± 0.01	−48.87 ± 0.19	47.99 ± 1.11
	Re-dispersed	232.40 ± 11.76	0.46 ± 0.05	−45.37 ± 0.91	25.27 ± 2.42
Z/S-cur	Freshly	81.30 ± 0.33	0.22 ± 0.01	−39.23 ± 0.23	81.50 ± 0.76
	Re-dispersed	88.40 ± 0.91	0.28 ± 0.01	−38.73 ± 0.33	73.83 ± 0.60
Z/S/C-cur	Freshly	70.40 ± 0.30	0.21 ± 0.02	−51.17 ± 0.16	90.98 ± 0.17
	Re-dispersed	72.50 ± 0.48	0.23 ± 0.01	−51.33 ± 0.62	89.20 ± 0.61

## Data Availability

Data is contained within the article or Appendix A.

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
