# Peer review of "Construction of a Ternary Composite Colloidal Structure of Zein/Soy Protein Isolate/Sodium Carboxymethyl Cellulose to Deliver Curcumin and Improve Its Bioavailability"

_foods, 2023, doi:10.3390/foods12142692_

Round 1

Reviewer 1 Report

The article was well written, and the topic is relevant and current, despite the results bringing incremental information without much innovation. The abstract is well-written and presents the work clearly. The introduction is adequate and brings current references on the subject. The material and methods section was well written, and the design of the experiments was well planned. The results are well described, and I suggest adding some references (doi: 10.3390/pr10122737 and 10.1016/j.foodhyd.2023.108638) to improve the discussion.

Reviewer 2 Report

The manuscript has investigated the possible application of ternary composite colloidal systems (zein/soy protein isolate/sodium carboxymethyl cellulose) to deliver curcumin and improve its bioavailability. The topic is interesting; However, the manuscript has several problems:

1. L 12; It is better to use "Z" instead of "zein".

2. Express more basic information about protein‐based colloidal systems; doi.org/10.1002/leg3.185.

3. L 260-263; provide the related information (as a Figure or Table) in the supplementary file.

4. Mention the significant letters for Tables and Figures.

5. Place Figures 3, 4, 5, and 6, and Table 2 after the related texts.

6. L 403; PI?

7. For the XRD experiment it is better to evaluate relative crystallinity.

L 336, 337; "However, the particle .... of CMC-Na", change this sentence.

Reviewer 3 Report

All detailed comments are included in the file next to the yellow ticks.

Round 2

Reviewer 3 Report

Now is better